# A Proposal for Modification of Plasmaspheric Electron Density Profiles Using Characteristics of Lightning Whistlers

Desy Purnami Singgih Putri [1], Yoshiya Kasahara [1,*], Mamoru Ota [2], Shoya Matsuda [1], Fuminori Tsuchiya [3], Atsushi Kumamoto [3], Ayako Matsuoka [4] and Yoshizumi Miyoshi [5]

1 Graduate School of Natural Science and Technology, Kanazawa University, Kakuma-machi, Kanazawa 920-1192, Japan
2 Department of Electrical and Control Systems Engineering, National Institute of Technology, Toyama College, Toyama 939-8630, Japan
3 Graduate School of Science, Tohoku University, Aoba-ku, Sendai 980-8578, Japan
4 Graduate School of Science, Kyoto University, Sakyo-ku, Kyoto 606-8502, Japan
5 ISEE, Nagoya University, Nagoya, Aichi 464-8601, Japan
* Correspondence: kasahara@staff.kanazawa-u.ac.jp; Tel.: +81-76-234-4952

**Abstract:** Reconstruction of reliable plasmaspheric electron density profiles is important for understanding physical processes in the plasmasphere. This paper proposes a technique that can be applied to correct the plasmaspheric electron density profiles using ray tracing by scrutinizing dispersion analyses of lightning whistlers. The Global Core Plasma Model and the International Reference Ionosphere were introduced as a reference electron density profile. Modifications of this electron density profile were then proposed to satisfy the dispersion characteristics of lightning whistlers measured by satellites in the plasmasphere. We first introduce two kinds of functions to modify the electron density: constant and linear, the linear function is more adequate. We applied our method to two lightning whistler events on 14 August 2017, measured by the Plasma Wave Experiment/Waveform Capture aboard the Arase satellite, and analyzed the dispersion of the observed lightning whistlers. We show how the density modification affects the delay time of the ray path and satisfies the dispersion characteristics under the appropriate adjustments.

**Keywords:** electron density; lightning whistler; plasmasphere; ray tracing; dispersion

## 1. Introduction

The plasmasphere is located above the ionosphere in Earth's magnetosphere. It is essential to Earth's space weather and is very sensitive to solar and geomagnetic conditions [1]. In recent decades, it has been challenging to research the behavior of plasmasphere conditions because measurements at this altitude are limited to satellite coverage, the conditions change daily, and it is strongly influenced by solar weather and geomagnetic conditions. Conventional ground station sounding techniques have been extended to topside sounding by placing sounders in orbit. The first topside sounder, Alouette I, an ionosonde, was launched in 1962. It sounded downward to determine the height distribution of the electron number density to the given location in the ionosphere [2]. The delay time of the signals from global navigation satellite systems (GNSSs), such as the Global Positioning System (GPS), provides total electron content (TEC) along the ray path from the GNSS satellites to the ground stations [3]. The Ionosphere Working Group of the International GNSS Service (IGS), created in 1998, was proposed to generate a reliable vertical TEC map [4].

Spacecraft observations can only measure in situ electron density along trajectories. Multicomponent instruments, such as triaxial magnetic coil sensors and electric antennas, often perform these measurements. These instruments measure the upper hybrid resonance frequency, which can provide electron density since the local magnetic field is known [5]. As an alternative to the in situ electron density measurement, remote sensing methods can

be employed as a measurement far away from observation points. One of these remote sensing techniques is radio sounding used by the Imager for Magnetopause-to-Aurora Global Exploration (IMAGE) satellite [6,7]. The radio plasma sounder emitted pulses over a range of frequencies and listened for returned echoes. By measuring the delay time, frequency, and direction of the echoes, the location and characteristics of the plasma at the remote reflection points could be derived [8]. This satellite worked well until 2005 when NASA announced they had lost communication with it. Goto et al. [9] proposed a method for deriving a global electron density model using the propagation characteristics of Omega signals observed by the Akebono satellite. Ray tracing calculates an Omega signal's wave normal directions and delays times under an appropriate electron density profile. The electron density profile can then be derived by fitting the calculation direction and time to the observed value. However, after the transmission of Omega signals was terminated, it became necessary to develop other methods for reconstructing global electron density profiles in the magnetosphere. Accordingly, we studied a new remote sensing technique aimed at deriving global electron density profiles from lightning whistlers and natural plasma waves detected in the magnetosphere instead of artificial signals.

A lightning whistler wave is an electromagnetic wave in the very low frequency (VLF) range originating from a lightning flash. It propagates in the geospace through the ionosphere. At the source point, the lightning whistler is simultaneously emitted in the 1–10 kHz frequency range. The wave generally propagates along Earth's magnetic field lines, as shown in Figure 1. After propagating and reaching the opposite hemisphere, a portion of the signal propagates down into the ionosphere. It can be detected on the ground, and some portion of the wave is reflected back to the ionosphere reaching the original hemisphere [10,11]. The propagation velocity is larger in the higher frequency range and becomes slower in lower frequencies, resulting in a dispersive spectrum, generally defined as dispersion. The propagation velocity depends on the electron density along the propagation path; therefore, the dispersive property strongly depends on the length of the propagation path and the TEC along the path.

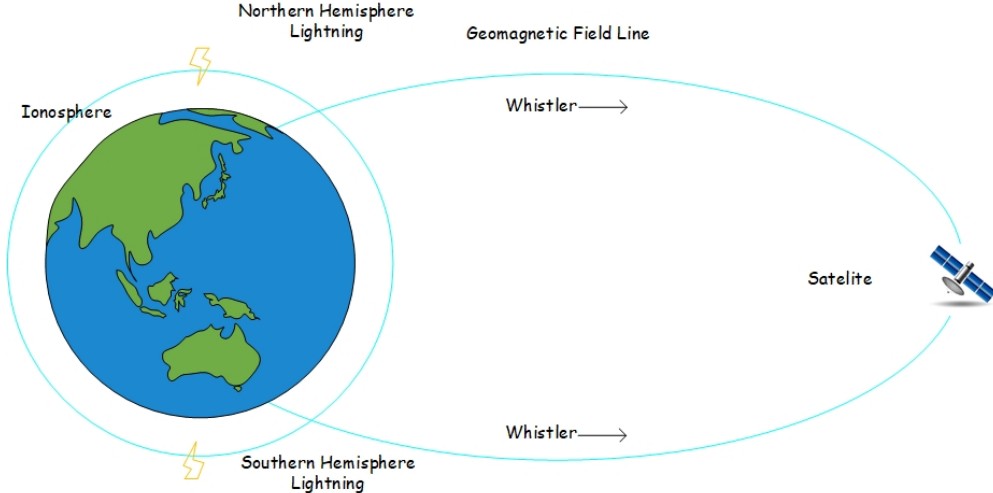

**Figure 1.** Propagation of lightning whistlers along the magnetic field line.

Lightning whistlers are often observed by spacecraft in the magnetosphere. Oike et al. [12] conducted a statistical analysis by examining the spatial and temporal distributions of lightning whistler frequencies observed by the Akebono satellite by automatically detecting their dispersion characteristics. Lightning whistlers were determined by deriving the gradient of a straight line plotted on a spectra diagram, with time $t$ on the horizontal axis and $\frac{1}{\sqrt{f}}$ on the vertical axis, where $f$ was the frequency of the lightning whistler. Umar et al. [13] also proposed a detection application and a classification system to detect lightning whistler waves observed by Arase. They investigated lightning whistlers using

image processing, pattern recognition, and classification. They used Bresenham's algorithm to detect line groups on time frequency diagrams to determine the presence of lightning whistlers. They then classified the detected lightning whistlers into six types based on shape and duration.

Bayupati et al. [14] analyzed two specific events representing the clear dispersion characteristics of lightning whistlers along the trajectory of the Akebono satellite. They compared the observed whistler dispersion trends with theoretically derived trends using a dipole geomagnetic field model and electron density profiles. They demonstrated that the observed trends mostly agreed with the theoretical results, and the optimization of the electron density profile could fit those small deviations. It showed that the dispersion analysis of lightning whistlers is a useful technique for reconstructing electron density profiles in the magnetosphere.

Santolik et al. [15] used a diffusive equilibrium (DE) model for plasma density distribution, using the measurements of intense whistlers observed by Arase and Van Allen Probes. They suggested adjustments to the electron density profile were also needed because of the differing values of the density simulation and the onboard observations.

Our study proposes a method that can be applied to reconstructing the plasmaspheric electron density profile using a ray tracing program. We demonstrate the reconstruction of the electron density profile along the ray path of the lightning whistler wave. We compared a lightning whistler spectrum reconstructed by ray tracing with the observed one to derive a more accurate electron density profile because the electron density profile computed from the International Reference Ionosphere (IRI) and Global Core Plasma Model (GCPM) does not always match with the electron density at the observation points, especially in the higher altitude region. We proposed modification functions to reconstruct a more adequate electron density profile. In the next section, we describe ray tracing and the models of electron density and magnetic field profiles used for ray tracing, and the basic characteristics of lighting whistler. In the Section 3, we present an overview of our simulation system and propose a modification function for the electron density profile. In the Section 4, we demonstrate how this method works followed by Section 5, and Section 6.

## 2. Fundamental Method

### 2.1. Ray Tracing

Lightning whistler waves are an important natural phenomenon that can provide valuable information about the Earth's magnetic field, ionosphere, and plasmasphere. They generally tend to follow the Earth's magnetic field lines, but they do not exactly propagate along the geomagnetic field lines and slightly deviate from the magnetic field lines. The lightning whistlers propagate with different delay times as a function of frequency and this feature is the most important point to derive the plausible electron density using ray tracing. By tracing the paths of individual rays, we can calculate quantitative propagation time (delay time) as a function of frequency, and obtain more complete and accurate features of the ray paths depending on the electron density profile.

Hazelgrove introduced the application of Hamilton's equations for a ray path in three dimensions. A ray path in a known magnetic field and plasma distribution can be represented by differential equations as follows [16]:

$$
\begin{aligned}
\frac{d\boldsymbol{r}}{d\tau} &= \frac{\partial D}{\partial \boldsymbol{k}}, \\
\frac{d\boldsymbol{k}}{d\tau} &= -\frac{\partial D}{\partial \boldsymbol{r}},
\end{aligned}
\tag{1}
$$

where $D$ represents the dispersion relation of the plasma wave, and $D(t, \boldsymbol{r}, \omega, \boldsymbol{k})$ is a constant along the ray path, where $t$ is the propagation time, $\boldsymbol{r}$ is the coordinate vector on the ray path, $\boldsymbol{k}$ is the wave normal vector, $\omega$ is the wave's angular frequency, and $\tau$ is the phase propagation time. In this study, the ray path is calculated in the coordinate system presented by $(r, \theta, \phi)$, where $r$, $\theta$, and $\phi$ are the geocentric distance, geomagnetic colatitude, and geomagnetic longitude, respectively. $\boldsymbol{k}$ is represented by $k_r, k_\theta, k_\phi$, three components of

the wave normal angle along the $r, \theta, \phi$ directions, respectively. The wave normal vector $\boldsymbol{k}$ along the ray path is provided by calculating the refractive index $\mu$, as the wave normal vector is proportional to the refractive index vector. Thus, we can numerically trace the ray path by solving the differential equations represented by Equation (1), and we regard this technique as ray tracing.

In a cold plasma approximation, the refractive index $\mu$ is obtained by solving Equation (2),

$$A\mu^4 + B\mu^2 + C = 0, \tag{2}$$

where $A$, $B$, and $C$ are functions of the normal wave angle, plasma frequency, and cyclotron frequency, respectively [17]. Cyclotron frequency and plasma frequency are determined by the function of the field intensity of the static magnetic field and the electron density, respectively. Therefore, appropriate magnetic field and electron density models are required to obtain propagation paths. Thus, adequate electron density profiles can be expected if plausible propagation paths are found that satisfy the observation data by way of the ray tracing program.

### 2.2. Magnetic Field and Electron Density Model

To use the ray tracing method, the magnetic field profile and the electron density profile must first be known. The simplest model for the magnetic field is the dipole magnetic field model. This model approximates the background magnetic field as a magnetic dipole at the center of the Earth. However, this work uses the International Geomagnetic Reference Field (IGRF) and the Tsyganenko model. The IGRF model, established by the International Association of Geomagnetism and Aeronomy, consists of mathematical models describing the internal part of Earth's magnetic field since 1900 A.D [18]. The Tsyganenko model is a semi-empirical magnetic field representation based on numerous satellite observations [19]. Interplanetary magnetic field parameters calculated from NASA's OMNI solar wind database were used to represent the magnetic field profile and were used in the ray tracing program.

In the previous research, Goto et al. [9] used the simplest electron density model, the DE model, as a reference of electron density profiles. This model assumes that the movement of protons, helium ions, and oxygen ions in plasma is in equilibrium with gravity along magnetic lines [20]. In contrast, the current research introduced the electron density profile by combining the GCPM and the IRI. The GCPM provides derived core plasma density as a function of geomagnetic and solar conditions. The plasmasphere model, as proposed by Gallagher et al. [21], only extends inward until $2R_E$, where $R_E$ denotes the radius of the Earth, and the IRI can be used up $\sim 600$ km, so the altitude range between them was calculated to complete the model. The IRI is a joint project between the Committee on Space Research and the International Union of Radio Science aimed at developing and improving an international standard for Earth's ionospheric parameters [22].

### 2.3. Dispersion of Lightning Whistlers

Observation satellites are equipped with instruments that measure components of the electric and magnetic fields. Measured lightning whistlers were investigated using a magnetic field component dataset. We used spectrograms, or dynamic power spectra, that showed variations in frequency $f$ and time $t$ to understand the dispersion characteristics of the lightning whistlers.

Eckersley's law [23] provides a simple relationship between the $f$ and $t$ of a lightning whistler as follows:

$$t = \frac{D}{\sqrt{f}} + t_0 \tag{3}$$

where $D$ is the dispersion of the lightning whistler and $t_0$ is the time of the lightning strike. Thus, the spectral form of the lightning whistler is shown in Figure 2a. When the dynamic

power spectra are graphed with $t$ on the horizontal axis and $\frac{1}{\sqrt{f}}$ on the vertical axis, as shown in Figure 2b, dispersion $D$ is determined by a straight line and calculated using the following equation:

$$D = \frac{t_2 - t_1}{\frac{1}{\sqrt{f_2}} - \frac{1}{\sqrt{f_1}}} \tag{4}$$

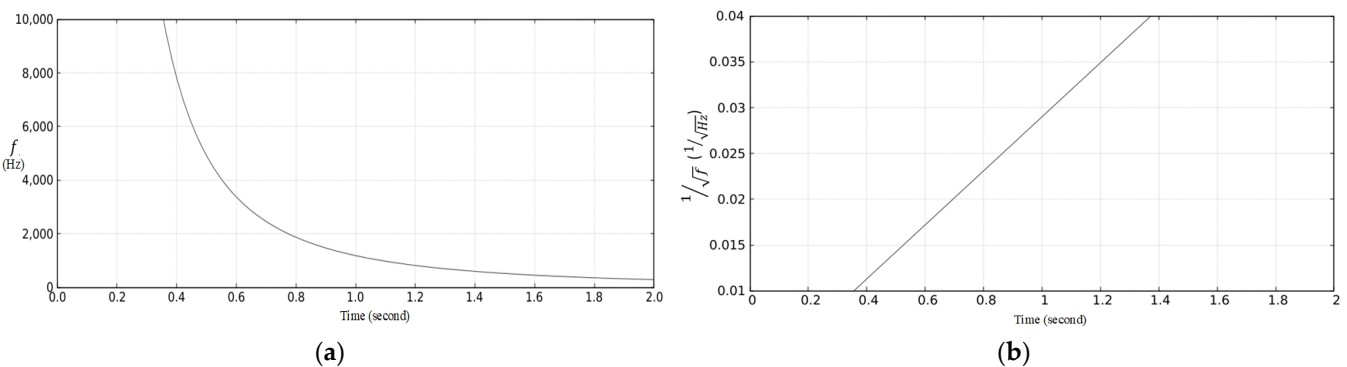

(**a**) (**b**)

**Figure 2.** (**a**) Dynamic power spectra of a lightning whistler. (**b**) Converted dynamic power spectra of a lightning whistler.

Using Equation (4), we calculated the dispersion of a lightning whistler using the ray tracing process for comparison with the observation.

Lightning sferics propagate around the world and stimulate the ionosphere to produce whistlers at every radius from the lightning stroke. They are dominant in the nearest 1000 km or so. We chose a region near a lightning stroke and a satellite footprint as the initial points for the ray tracing simulation. To obtain the location of the lightning stroke, we referred to the World Wide Lightning Location Network (WWLLN) database. It locates lightning globally using sparsely distributed VLF detection stations [24].

The main point of the paper is to propose a method using the propagation characteristics of lightning whistlers. As we describe in Section 2.1, lightning whistlers propagate with different delay times as a function of frequency. Thus, if we can find an electron density profile by which the propagation delay obtained by ray tracing agrees with the observed one at all frequencies, we can say that the derived electron density profile is fairly reliable. Furthermore, it is also important to stress that our proposed method is capable of estimating the electron density profile by reproducing the relative delay time (what we call "dispersion $D$") of the lightning whistler, even if some offset is included in the lightning occurrence time obtained by WWLLN and/or the observation time at the satellite.

## 3. Proposed Method

### 3.1. Calculation Techniques

There are fewer observation data in the plasmasphere compared with a lower altitude in the ionosphere. Therefore, the electron density model is sometimes deviated from the in situ electron density measured by the satellites. Consequently, electron density profiles require some adjustment. We developed the original ray tracing program introduced by Kimura and Goto [20] and Goto et al. [9]. We extracted the module that calculates the electron density profile referring to IRI and GCPM from the original raytracing to simplify the adjustment process. An overview of the ray tracing simulation process is shown in Figure 3.

After producing the electron density profile (0) using the GCPM and IRI model, as shown in Figure 3, we modified and generated electron density profiles, from profiles (1) to ($n$), to seek the adequate electron density profiles among them. We also calculated the first derivative of $\mu$ to solve Equation (1) for ray tracing. Therefore, the electron density profiles (0) to ($n$) comprise the absolute density and its first partial derivatives in the

$(r, \theta, \phi)$ coordinate system. In this work, we used $10°$ as a resolution of longitudinal, $1°$ for latitude, and $R_E/10$ for altitude. It took more than three hours to complete this process, and if we use a smaller resolution, it can take longer. We then calculated the ray paths of whistler mode wave corresponding to the electron density profiles $(0 - n)$ and selected the best-fit profile to meet the observed spectrum of lightning whistlers for all frequency ranges (1–10 kHz) with the one theoretically reconstructed by ray tracing calculating the propagation time of lightning whistler.

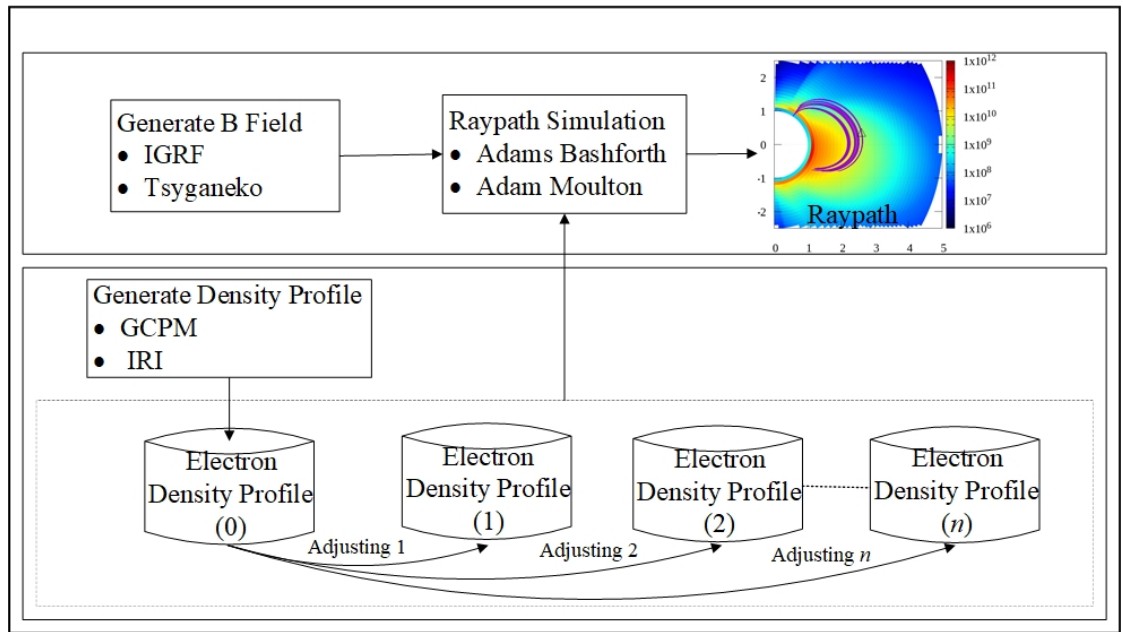

**Figure 3.** Flowchart of the ray tracing simulation system.

### 3.2. Proposed Modification Function

The modification function is generally represented by $f(r, \theta, \phi)$. In the current paper, however, we only applied $f(r)$, as a preliminary study to modify the derived density $N$ which became

$$\hat{N} = f(r)N \tag{5}$$

where $\hat{N}$ is the modified density and $r$ is altitude. The first simple function we used was the constant $f(r) = c$ to multiply the whole density. The constant $c$ is derived from the ratio of the observed density and that obtained from the density model. The second step was to retain the electron density profile for the lower altitude and use a linear equation to adjust the density, as shown in Figure 4, as follows:

$$f(r) = \begin{cases} 1 & R < r_0 \\ aR + b & r_0 \leq R \leq r_m \\ c & R > r_m \end{cases} \tag{6}$$

where $r_0$, $r_m$, and $c$ are constants that can be changed depending on the case. The first part of our modification function involves a constant value of 1, which is designed to maintain the density below $r_0$. In this paper, we set $r_0$ at 1000 km assuming that the IRI is reliable for the ionosphere. On the other hand, we determined $c$ constant in third part using the ratio of the in situ electron density measurement and the one from the reference electron density model. Finally, we tried to find the best $r_m$ from ray tracing simulation by changing the value of $r_m$ that fits the dispersion characteristics for easy determination of these unknown parameters. The graph of this modification function is shown in Figure 4. We noted that this modification causes discontinuity in the derivatives of the electron density profile, and it may affect the derivatives of refractive index at $r_0$ and $r_m$ in the ray

tracing calculation, but this effect is negligible because we calculated the derivatives by interpolating the neighboring grid data stored in the electron density model.

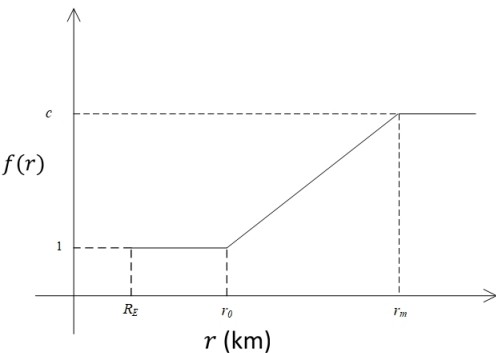

**Figure 4.** Graph of the proposed density modification using a linear function.

After adjusting the density, we performed a ray path simulation using the initial point near the lightning stroke location. We indicated some areas around the stroke as initial points for the ray tracing simulation.

## 4. Implementation

### 4.1. Observed Lightning Whistler

We used Waveform Capture (WFC) data [25], a subsystem of the Plasma Wave Experiment (PWE) instrument [26] onboard the Arase satellite [27]. The WFC is a waveform receiver that measures two electric field components up to 20 kHz and three magnetic field components up to 20 kHz. Lightning whistlers measured by WFC were investigated using the magnetic field dataset. Figure 5 shows the dynamic power spectra produced from WFC data [28] measured on 14 August 2017, from 08:31:51 to 08:31:55 UT, after this, referred to as the first event, generated by performing Fourier transform. As shown in Figure 5a, the lightning whistler propagated at greater velocities in the higher frequency range compared with the lower frequency range.

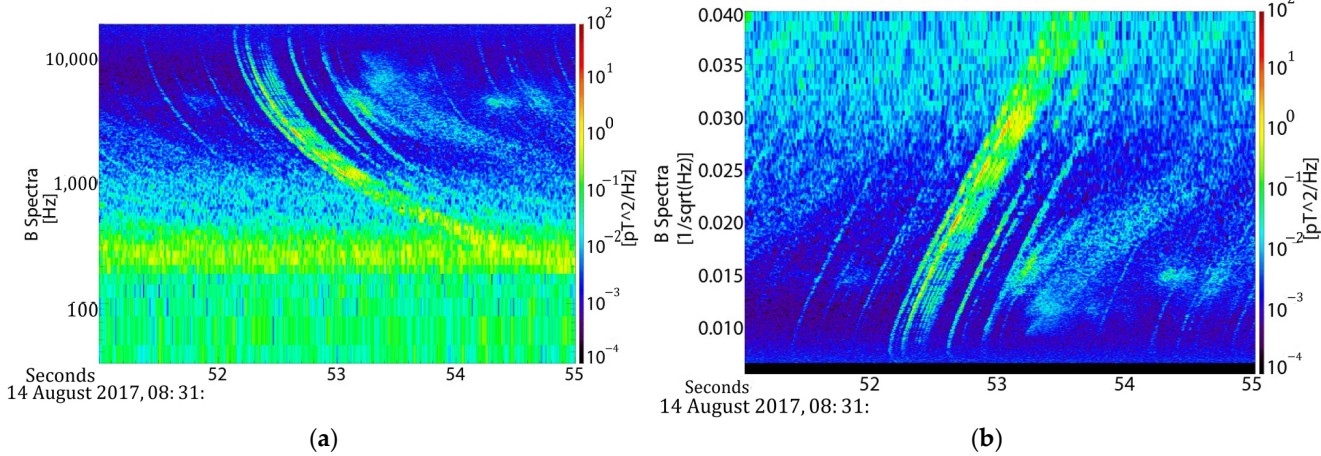

(**a**)                                                 (**b**)

**Figure 5.** Lightning whistler observed by the Arase satellite on 14 August 2017 at 08:31:51–08:31:55 UT: (**a**) Dynamic power spectra of lightning whistler waves. (**b**) Converted dynamic power spectra of lightning whistler waves.

Figure 5b shows the dynamic power spectra graphed with $t$ on the horizontal axis and $\frac{1}{\sqrt{f}}$ on the vertical axis. Using Equation (4), we calculated the dispersion of the lightning whistler detected by Arase as $\sim 29.6$.

We found the source lightning candidate from the WWLLN database at latitude $52.28°$ and longitude $119.783°$ at 08:31:51.874189 UT, located near Russia. At that time, the distance

between the Arase footprint and the lightning source point was around 667.3 km, as shown in Figure 6. The Arase footprint was taken using a magnetic field north trace with IGRF and Tsyganenko model [29].

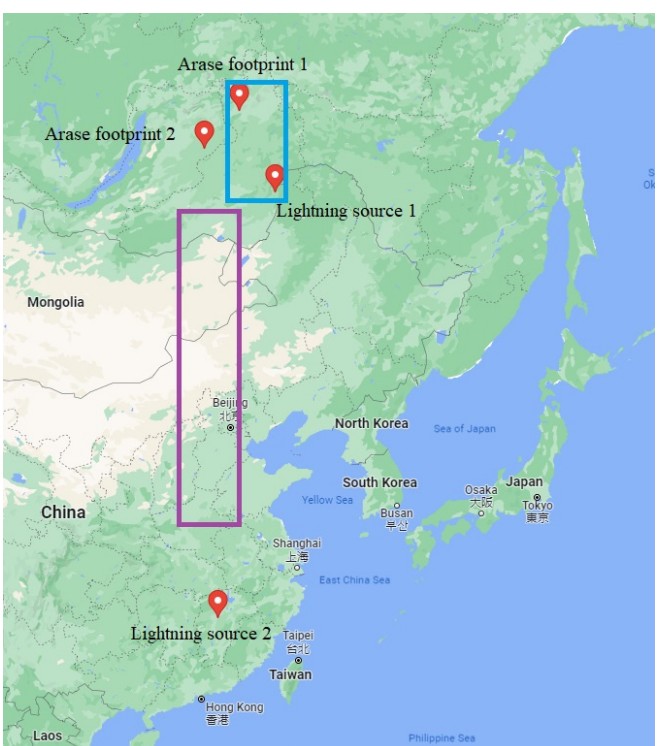

**Figure 6.** Position of the Arase footprint and the lightning source point.

The second event was taken from the same date at 08:28:06–08:28:10 UT (Figure 7a). From the converted dynamic power spectra (Figure 7b), the dispersion of the second event was ~30. Unlike the first event, the second event's lightning location is far from the Arase footprint, approximately 2900 km (Figure 6). Based on the WWLLN database, this event's lightning candidate is at latitude 27.94° and longitude 115.409° at 08:28:07.087262 UT.

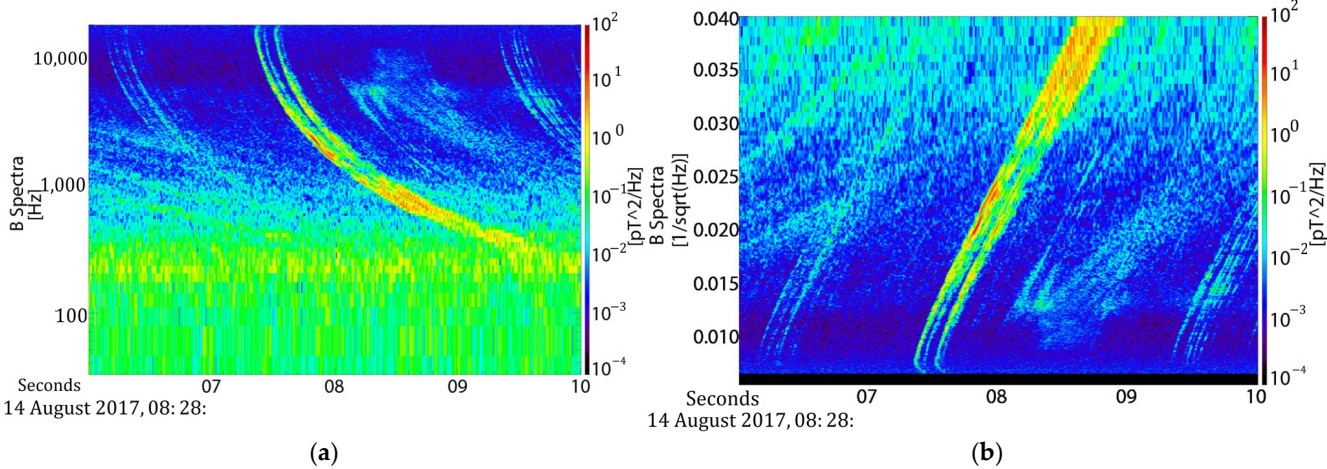

(**a**)

(**b**)

**Figure 7.** Lightning whistler observed by the Arase satellite on 14 August 2017 at 08:28:06–08:28:10 UT: (**a**) Dynamic power spectra of lightning whistler waves. (**b**) Converted dynamic power spectra of lightning whistler waves.

We noted that lightning discharges produce a wide range of electromagnetic waves as sferics and whistlers. The sferics can travel through the Earth–ionosphere waveguide over

long distances with its attenuation rate of only around 2–3 dB per 1000 km of propagation distance [30]. Therefore, it is feasible to create the ray path in this second event, despite the fact that the lightning source is further away from the Arase footprint.

*4.2. Result 1*

For the first event, we generated a reference electron density profile for 14 August 2017, at 08:31:52 UT with Kp index of 0. The two proposed modifications were made to determine how they affected the ray path in the ray tracing program. We used the following initial conditions as shown in Table 1 to see how the modification of the electron density profile affected the propagation characteristics of the lightning whistler. Since we know that the source point is in the northern hemisphere, we set the $k$ vector, the propagation direction angle, with $Del_{in}$ as the angle between wave normal and magnetic field and $Eps_{in}$ azimuthal wave normal angle, as written in Table 1. We ran the ray tracing program for several density modifications.

**Table 1.** Initial conditions of ray path for the first event.

| | |
|---|---|
| Frequency(kHz) | $1 - 10$ |
| Altitude $R$ | $(1 \sim 1.5)R_E$ |
| Longitude (degree) | $117.0 - 119.0$ |
| Latitude (degree) | $47.0 - 59.0$ |
| $k$ vector (degree) | $Del_{in} : 10°$ |
| | $Eps_{in} : -10° \sim 10°$ |

To determine the propagation time for each ray path, we assumed that the path passed the Arase with a distance of the path and the Arase location was less than 1000 km. The results are presented in Figure 8. The horizontal axis shows the propagation time, and the vertical axis shows the $\frac{1}{\sqrt{f}}$. Each point with light blue, orange, yellow, dark blue, and purple are the calculated propagation times from the source point to the Arase location in the 0–10 kHz frequency range using different electron density profiles, respectively.

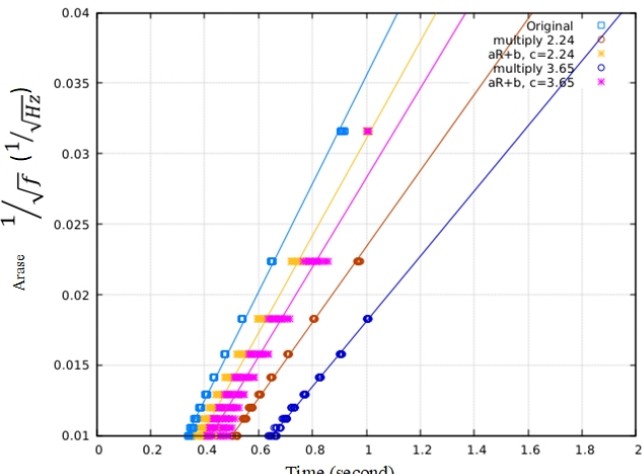

**Figure 8.** The calculated propagation time from the source point to the Arase location.

Changing the electron density profile affects the propagation time and dispersion, as shown in Figure 8. The colored lines indicate the dispersion for each calculated propagation time. The light blue line indicates the propagation time using density (0) or without modification. The orange and dark blue lines indicate the propagation times modified by $f(r) = c$ with $c$ as 2.24 and 3.65, respectively. Multiplying the density by a higher number slowed the propagation time and increased dispersion. Using Equation (5) to modify the density resulted in the yellow and purple lines, where $c = 2.24$ and $c = 3.65$

were used, respectively, with the same $r_0 = 1000$ km $+ R_E$ and $r_m = 2R_E$. The density at the lower altitude was retained to ensure that the propagation time was not too slow or that the dispersion was too large. Therefore, we can approach the desired dispersion characteristics by finding the appropriate free parameters in the modification functions and initial conditions.

To satisfy the electron density at the observation satellite point, we set the $c$ constant of Equation (6) using the ratio of the in situ electron density measurement from the Arase and from the reference electron density model at the same location. The electron density along the Arase trajectory can be derived from the upper hybrid resonance (UHR) frequency, measured using a high frequency analyzer (HFA) [31,32], a subsystem of the PWE. The DC magnetic field intensity is measured using the magnetic field experiment (MGF) [33]. By setting $r_0$ to 1000 km from Earth's surface, we then examined several ray paths changing the value of $r_m$ that fits the dispersion characteristics and obtain $r_m = 2R_E$. This result produced $c = 2.24$, $a = 2.3 \times 10^{-7}$, and $b = -0.7011$.

The density changes are shown in Figure 9. In Figure 9 there are three plots (a), (b), and (c) that show the reference electron density profile, modified electron density using $f(r) = c$, and the one modified by Equation (6), respectively. We generated the density for all regions, but Figure 9d shows for a particular slice at the geographic coordinate of longitude $120°$ and $9°$. The horizontal axis shows the altitude in km, and the vertical axis shows the electron density in/m$^3$ with logarithm scale. The blue, orange, and yellow colors describe the electron density (0) or without modification, the modified electron density with constant $c$, and with Equation (6) at each altitude point, respectively. The electron density profiles at higher altitudes are all similar because the electron density is small. The density at the Arase location is shown with a purple triangle, which is $\sim \frac{3800}{\text{cm}^3}$.

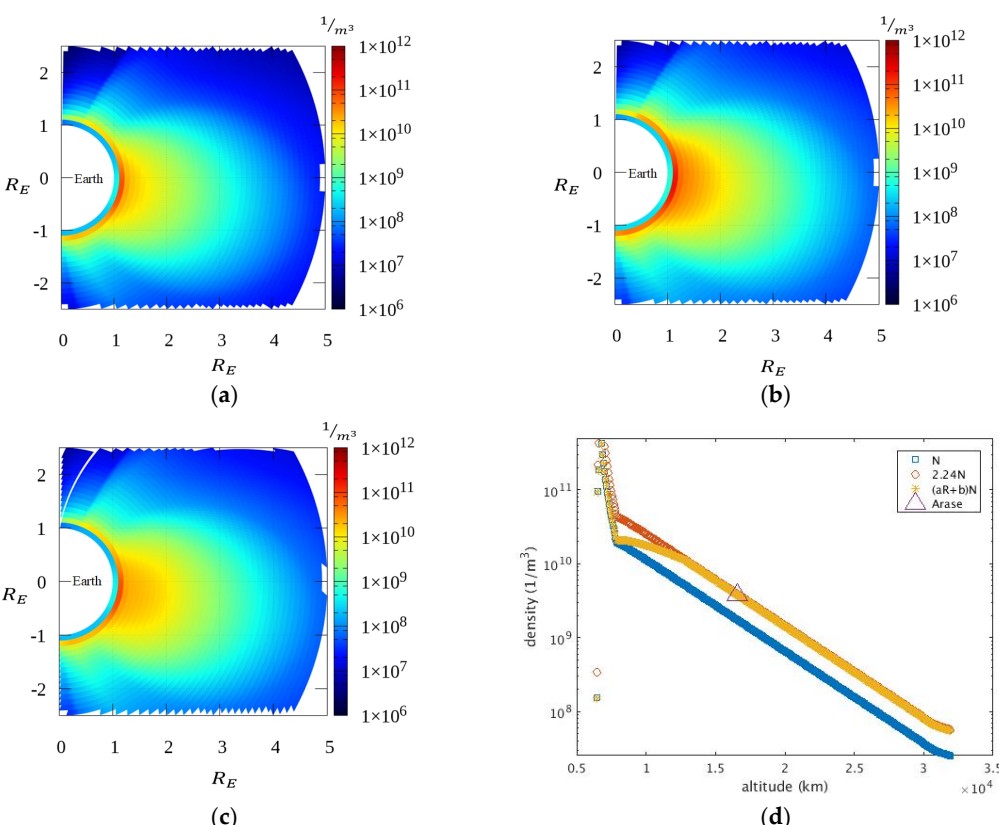

**Figure 9.** The generated electron density profiles at longitudinal $120°$: (**a**) without modification; (**b**) modified by $f(r) = c$; (**c**) modified by Function 6. (**d**) The reference electron density without and with modification at the geographic coordinate of longitude $120°$ and latitude $9°$.

We calculated the ray path using these electron density profiles for all frequency ranges. This simulation can provide information on the direction, intensity, and time of arrival of the rays at different points in three dimensions. To see how the ray path passes the satellite position, we plotted the ray path in two dimension for a particular slice at longitudinal 120° as shown in Figure 10a–c. This shows the electron density profile in color contours and the ray paths at 5 kHz, 8 kHz, and 10 kHz under the modified electron density profile using Equation (6). The purple line describes the ray paths, and the black triangle indicates the location of the Arase. The different frequencies of the ray result in slightly different paths.

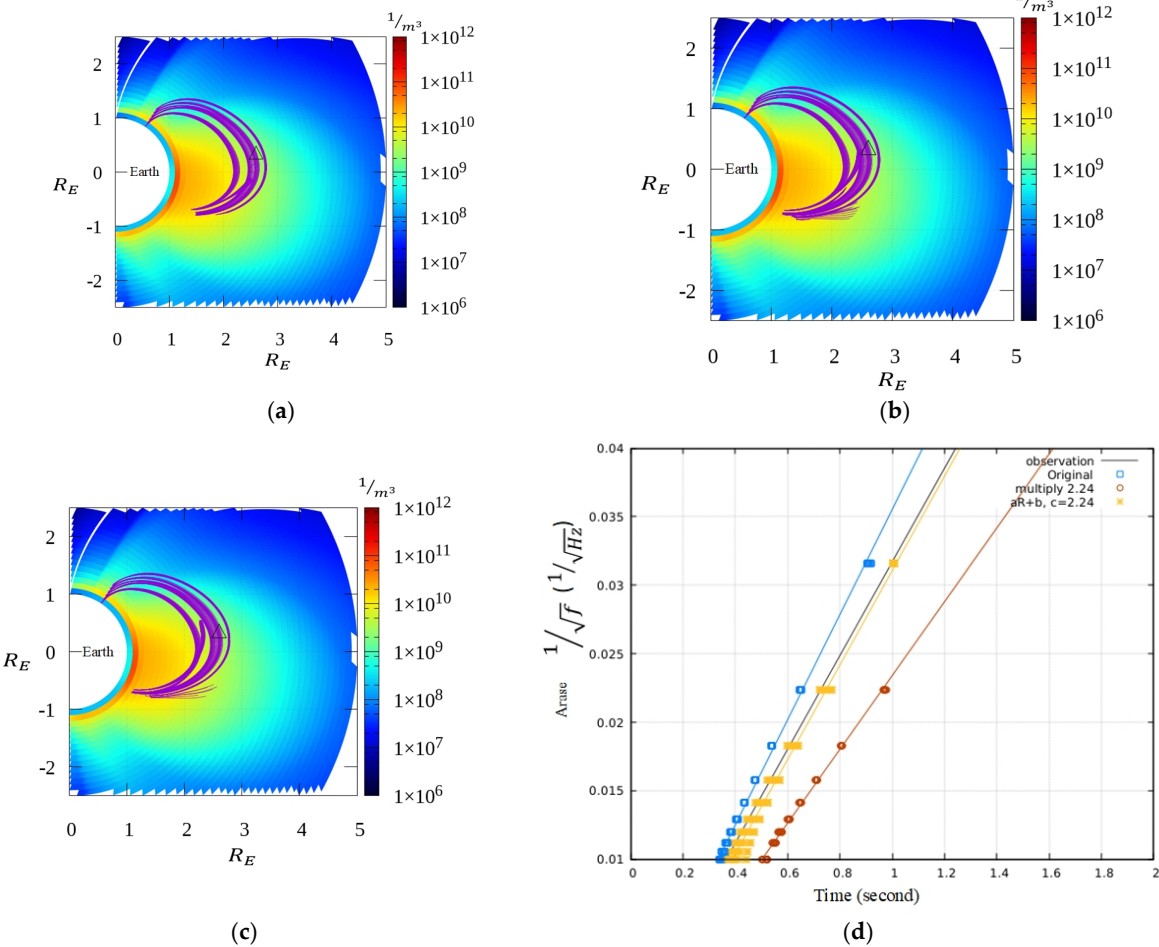

**Figure 10.** Simulation results. Ray path simulation using the density modified by Equation (6) for frequency: (**a**) 5 kHz, (**b**) 8 kHz, (**c**) 10 kHz. (**d**) The calculated propagation times from the source point with Arase location for the first event; gray line shows the dispersion measured by Arase; blue line shows the calculated dispersion using the reference electron density profile; orange line shows the calculated dispersion using the modified electron density with constant $c$, yellow line shows the calculated dispersion using the modified electron density using Equation (6).

We then calculated the dispersion for each condition, as shown in Figure 10d. The propagation time of the ray paths under the reference electron density profile are slightly faster than the observation time, shown with the blue line, and the difference in the propagation time was much larger in the lower frequency range. After multiplying the density by a constant $c$ on modification function $f(r) = c$, shown with an orange line, the propagation time was much slower and the dispersion much larger than the observed propagation time and dispersion, respectively. In addition, any propagation time for the lower frequency range ($< 2$ kHz) was identified as the ray paths passed away from the location of the Arase. The second adjustment, using Equation (6), resulted in a dispersion

similar to the observation ($\sim$ 29.06) and a propagation time at 10 kHz of 0.370238 s. The appropriate density modification met the propagation characteristics.

The second adjustment produced a time for the lightning derived from the ray tracing at 10 kHz around 08:31:51.879762 UT. It was only 5.5573 milliseconds different from the WWLLN data; therefore, the second adjustment using function (6) satisfied the propagation characteristics of the lightning whistler.

### 4.3. Result 2

We generated a reference electron density profile for the second event at 08:28:07 on the same date. Since the distance between the Arase footprint and lightning source location is quite far, we set the ray tracing simulation region between these locations (Figure 6) with a purple rectangle. We used the following initial conditions to run the simulation (Table 2).

**Table 2.** Initial conditions of ray path for the second event.

| Frequency (kHz) | $1 - 10$ |
|---|---|
| Altitude $R$ | $(1 \sim 1.5) R_E$ |
| Longitude (degree) | $116.0 - 117.0$ |
| Latitude (degree) | $35.0 - 50.0$ |
| $k$ vector (degree) | $Del_{in} : 10^{\circ}$ |
| | $Eps_{in} : -10^{\circ} \sim 10^{\circ}$ |

In this second simulation, we set the constant $c = 2.35$ from the density ratio of the in situ measurement and the one calculated from the reference electron density model. We also set the parameters $r_0 = 1000$ km $+ R_E$ and $r_m = 2R_E$, the same as in the first event. Table 3 presents the produced parameters. Even though parameter $c$ changes slightly due to the satellite position, the altitude ranges to be modified are the same in both cases. Then, we simulated the ray path for the electron density profile from 1 to 10 kHz and calculated the dispersion for each condition (Figure 11).

**Table 3.** Modification parameter for event 1 and event 2.

| Parameter | $c$ | $r_0$ | $r_m$ | $a$ | $b$ |
|---|---|---|---|---|---|
| Event 1 | 2.24 | 1000 km $+ R_E$ | $2R_E$ | $2.3 \times 10^{-7}$ | |
| Event 2 | 2.35 | 1000 km $+ R_E$ | $2R_E$ | $2.5102 \times 10^{-7}$ | $-0.8520$ |

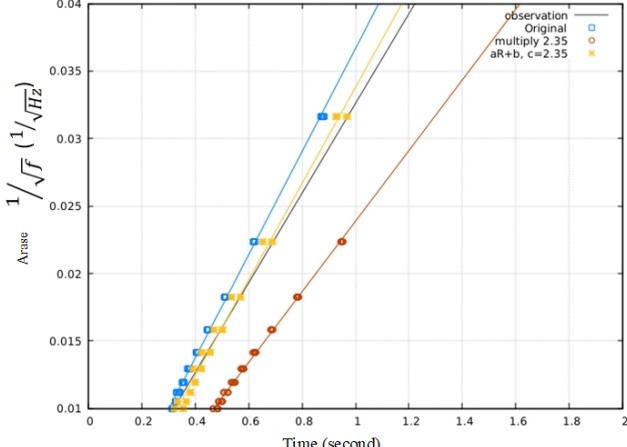

**Figure 11.** The calculated propagation times from the source point with the Arase location for the second event. The gray line shows the dispersion measured by Arase, the blue line shows the calculated dispersion using the reference electron density profile, the orange line shows the calculated dispersion using the modified electron density with constant $c$, and the yellow line shows the calculated dispersion using the modified electron density using Equation (6).

Similar to the first event, Figure 11 shows that the reference density without adjustment causes a faster propagation time and a smaller dispersion than those of the observation propagation data from Arase onboard. The first modification also provided a slower propagation time and a larger dispersion than the observation. The second adjustment produced a similar dispersion (~29.6) and propagation time to the observation. This adjustment resulted in the lightning occurrence time derived from the ray tracing at 10 kHz around 08:28:07.090979, with a time difference of approximately 3.717 ms compared with the data from WWLLN.

## 5. Discussion

In this study, we applied the proposed method to two lightning whistlers measured using the Arase satellite in the plasmasphere. In both events (Figure 7), the Arase footprints and the lightning source locations are longitudinally close. In the analyses, we first compared the electron densities generated from the IRI and GCPM models with the ones measured in the plasmasphere using the Arase and revealed a several-fold difference between them. We applied a modification function (Equations (5) and (6)) to modify the plasmaspheric electron density profile, assuming that the IRI is more reliable, as it is derived from extensive measurement data in the ionosphere. Next, we evaluated the effectiveness of our proposed method by estimating the dispersion characteristics of lightning whistlers. We obtained differences in the delay time less than 6 ms for both cases. The modification parameters remain the same ($r_0$ and $r_m$), whereas parameter $c$ differs in these two events, indicating that the electron density profiles for both events are adequately corrected toward the altitude.

Our analysis is limited in the narrow longitudinal region, and the derived modification parameters cannot be fully applied to the global electron density, especially longitudinally, as the propagation path is in the limited meridian plane. However, the results suggest that our method can be expanded longitudinally and latitudinally while using the measured electron data along the satellite trajectory. Continuous detection of lightning whistler waves can assist in modifying the plasmaspheric electron density profile. Therefore, our findings demonstrate the significant potential of this method to enhance our understanding of the global electron density distribution.

Comparing our results with the other models and TEC measurements is necessary to evaluate the proposed method's effectiveness and reliability. However, we must further examine and confirm the appropriate modification function, including the latitude and longitude, and thus, the quantitative accuracy evaluation is beyond this paper's scope.

## 6. Conclusions

We proposed a technique to correct the plasmaspheric electron density profiles using ray tracing via scrutinizing dispersion analyses of lightning whistlers. We demonstrated that modifying the electron density profile affects the dispersion trend of a lightning whistler wave in the plasmasphere. We applied our proposed method to two lightning whistler events in the plasmasphere detected by Arase and demonstrated that both events' electron density profiles are adequately corrected toward the altitude. Furthermore, they are plausibly corrected if we introduce the latitudinal and longitudinal modification functions along the satellite trajectory.

The modification parameters should be primarily represented as a function of altitude, latitude, and longitude, while we only applied a modification function in the altitude direction as a preliminary study. We first fixed the constant $r_0$ at 1000 km, relying on the electron density in the ionosphere provided by the IRI model, which should be derived from extensive measurement data. Next, we determined the constant $c$, comparing the in situ electron density derived from the Arase measurements. Finally, we examined several ray paths changing the value of $r_m$ that fits the dispersion characteristics of lightning whistlers. In addition, we manually investigated the appropriate $r_m$ values in this paper, but it is necessary to run this process using automatic fitting. Further study is necessary

to examine and confirm the appropriate modification function, including the latitude and longitude, and compare our results with the other models and the TEC measurements to evaluate the proposed method's effectiveness and reliability.

**Author Contributions:** D.P.S.P. developed the proposed method to modify the reference electron density profiles using ray tracing and wrote the paper. Y.K. is a supervisor of this research project and a principal investigator of the PWE. M.O. developed the original ray tracing program. S.M., the Co-Is of the PWE, was responsible for the development of the onboard software aboard Arase/PWE. F.T. and A.K., as Co-Is of the PWE, were responsible for the development of the HFA. A.M. is the PI of the MGF and was responsible for the deployment of the MSAT and WPT. Y.M. is the project scientist of the ERG (Arase) satellite. All authors have read and agreed to the published version of the manuscript.

**Funding:** We sincerely thank Indonesia's Directorate General of Higher Education, Research, and Technology for its financial support and Indonesia's Ministry of Education Culture, Research, and Technology for supporting the fund. This research was partially supported by Grant-in-Aid for Scientific Research from the Japan Society for the Promotion of Science (21H01146 and 22K19845).

**Data Availability Statement:** Science data of the ERG (Arase) satellite were obtained from the ERG Science Center [34] operated by ISAS/JAXA and ISEE/Nagoya University (https://ergsc.isee.nagoya-u.ac.jp/index.shtml.en, accessed on 1 June 2022) Miyoshi, Hori et al., 2018.

**Conflicts of Interest:** The authors declare no conflict of interest.

## Abbreviations

| | |
|---|---|
| VLF | Very Low Frequency |
| TEC | Total Electron Content |
| GCPM | Global Core Plasma Model |
| IRI | International Reference Ionosphere |
| IGRF | International Geomagnetic Reference Filed |

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
