# Peer review of "A Proposal for Modification of Plasmaspheric Electron Density Profiles Using Characteristics of Lightning Whistlers"

_remotesensing, doi:10.3390/rs15051306_

Round 1

Reviewer 1 Report (Previous Reviewer 1)

The authors have revised properly throughout the manuscript.

Author Response

We express our genuine appreciation to the reviewer for their valuable insights and productive recommendations after thoroughly reviewing our manuscript. We have revised the manuscript remotesensing- 2210927 entitled A proposal for modification of plasmaspheric electron density profiles using characteristics of lightning whistlers. We are sincerely hopeful that our revised manuscript will be received positively and accepted for publication in Remote Sensing.

Sincerely yours,

Desy Purnami Singgih Putri

Reviewer 2 Report (Previous Reviewer 3)

The electron density profile modification method proposed in the paper is not physically justified. Yes, we can take electron density profile and modify like it is proposed in the paper, to match signal delay time. We can also take different modifier to the profile and achieve same delay. However it does mean that profile is correct. Please provide reasoning why modification consist of three parts and why they are what they are (constant, linear and again constant).

The applicability of ray tracing is not clear. Lightning whistlers propagate along geomagnetic field lines (Figure 1 shows the same). There is no need of the ray-tracing in that case, since the IGRF already give correct trajectory of the whistler wave.

The modification method cause discontinuity for r0 and rm which manifest it self in Figure 9 (first yellow star jumps compared to the second).

Modification depends only on height (assuming r is radial direction), but ray-tracing is made in 3D or at least 2D media (according to Figure 10). How the modification applied here?

No discussion is provided on how electron density profile in the magnetosphere should look like. Proposed method is based on the modification of existing electron density profile as defined by equations (6). The electron density profiles is derived from GCPM and IRI. There many researches that study empirical models of the plasmaspheric electron density. They provide simple formula that could be modified without equations (6) which physical justification is not clear.

Second event has lightning source much farther from Arose footprint, however has greater whistler amplitude compared to the first event (according to comparison Figures 5 and 7).

Authors doesn’t introduce the approach to understand how the results could be validated. Only two events are considered and no comparison with experimental result or models are made. There is no enough statistics to see modification parameters distribution.

Ray tracing is complicated procedure, while equation (1) and (2) give basis for it, the software implementation is not that simply. Please provide the reference to the code you are using for the ray tracing or reference to previous works where the implementation and validation is described.

Author Response

We express our genuine appreciation to the reviewer for their valuable insights and productive recommendations after thoroughly reviewing our manuscript. We have revised the manuscript remotesensing- 2210927 entitled A proposal for modification of plasmaspheric electron density profiles using characteristics of lightning whistlers. We are sincerely hopeful that our revised manuscript will be received positively and accepted for publication in Remote Sensing.

We show the revised part in the manuscript with track changes.

Sincerely yours,

Desy Purnami Singgih Putri

Point 1

The electron density profile modification method proposed in the paper is not physically justified. Yes, we can take electron density profile and modify like it is proposed in the paper, to match signal delay time. We can also take different modifier to the profile and achieve same delay. However it does mean that profile is correct. Please provide reasoning why modification consist of three parts and why they are what they are (constant, linear and again constant).

Response 1

Thank you for your comment. The main point of the paper is to propose a method using the propagation characteristics of lightning whistlers. Lightning whistlers propagate with different delay times as a function of frequency. Thus, if we can find an electron density profile by which the propagation delay obtained by ray tracing agrees with the observed result at all frequencies, we can say that the derived electron density profile is fairly reliable.

Furthermore, it is also important to stress that our proposed method is capable of estimating the electron density profile by reproducing the relative delay time (what we call “dispersion”) of the lightning whistler, even if some offset is included in the lightning occurrence time obtained by WWLLN and/or the observation time at the satellite.

We realize that such description is not shown in the previous revision and we added it in the current revision in lines 204-213

The idea of Function (6) is to modify the electron density above altitude r0. The first part of our modification function involves a constant value of 1, which is designed to maintain the density below r0 (in this paper, we use 1,000km assuming that the IRI is reliable for the ionosphere (as described in lines 302-303 in the previous revision and in lines 247-250 in the current revision). On the other hand, we can determine  constant in Function (6) using the ratio of the in situ electron density measurement and the one from the reference electron density model. Finally, we try to find the best rm from ray tracing changing the value of rm that fits the dispersion characteristics for easy determination of these unknown parameters.

This description can be found in subsection 3.2 (Proposed modification function)  in lines 247-254 in the current revision.

Next, as was pointed out by reviewer #2, we agree that the modification Function (6) is not a unique solution and other solutions might be possible. But we applied this Function (6) as a preliminary study in the present paper, because it is easy to determine unknown parameters in it in the following process.

Point 2

The applicability of ray tracing is not clear. Lightning whistlers propagate along geomagnetic field lines (Figure 1 shows the same). There is no need of the ray-tracing in that case, since the IGRF already give correct trajectory of the whistler wave.

Response 2

Lightning whistler waves are an important natural phenomenon that can provide valuable information about the Earth's magnetic field, ionosphere, and plasmasphere. Despite their tendency to follow the Earth's magnetic field lines, they do not exactly propagate along the geomagnetic field line, but slightly deviate from the magnetic field line. The lightning whistlers propagate with different delay times as a function of frequency and this feature is the most important point to derive the plausible electron density using ray tracing. By tracing the path of individual rays of the signal through the Earth's magnetic field, ionosphere, and plasmasphere, taking into account factors such as the density of these regions and the frequency of the signal, we can gain a comprehensive understanding of the behavior and interactions of lightning whistler waves with the Earth's environment. By using this method we could get a more complete and accurate picture of the path and behavior of lightning whistlers, including the effects of changes in the ionosphere's density along the way.

We realized that this description is essential to show the significance of our proposed method, so that we added them in subsection 2.1 (Ray tracing) lines 123-131.

We also added more plots in Figure 10 to show that the ray propagates differently depending on the frequency.

Point 3

The modification method cause discontinuity for r0 and rm which manifest it self in Figure 9 (first yellow star jumps compared to the second).

Response 3

The jump of the yellow star seen in Figure 9 is due to the bad resolution of the profiles in the figure, so we replaced the figure with fine resolution. We also changed the vertical axis in logarithm to show the electron density profiles more clearly.

On the other hand, we agree that discontinuity appears in the derivatives of the electron density profile, and it may affect the derivatives of refractive index at r0 and rm in the ray tracing calculation, but this effect is negligible because we calculate the derivatives by interpolating the neighboring grid data stored in the electron density model. A more detailed description is shown in Response 8 (in the calculation process of ray tracing.)

Point 4

Modification depends only on height (assuming r is radial direction), but ray-tracing is made in 3D or at least 2D media (according to Figure 10). How the modification applied here?

Response 4

The electron density profile is generated for all regions, from the Earth's surface up to 5 RE (the radius of the Earth). We added 3 plot graphs in Figure 9 (a), (b), and (c) to show the electron density profiles for a particular slice at longitude 120o.

While Figure 9 (d) shows the reference electron density without and with modification at longitude 120o and latitude 9o to view the electron density modification, especially at the Arase location.

Our ray tracing program can calculate the ray path in 3D. However, we plotted the ray path in 2D at longitudinal 120o to see how the ray path passes the Arase as shown in Figure 10 (a).  

Point 5

No discussion is provided on in how electron density profile in the magnetosphere should look like. Proposed method is based on the modification of existing electron density profile as defined by equations (6). The electron density profiles is derived from GCPM and IRI. There many researches that study empirical models of the plasmaspheric electron density. They provide simple formula that could be modified without equations (6) which physical justification is not clear.

Response 5

As we replied in Response 1, we agree that the proposed modification, Equation (6), is not a unique solution and other solutions might be possible. But we applied this as a preliminary study, because it is easy to determine unknown parameters as described in Response 1.

Nevertheless, we stress again that the main point of the paper is to propose a method using the propagation characteristics of lightning whistlers using its wide frequency range, and our analysis shows that we can meet lightning characteristics with less than 6 ms delay time difference.

We agree that our current study is limited to a narrow longitudinal region, and the derived parameters cannot be applied to the global electron density, especially longitudinally. In that sense, we need to try to find more meaningful modification functions in future work.

But, we can expand the method by using measured electron data along the satellite trajectory. Continuous detection of lightning whistler waves can assist in modifying the plasmaspheric electron density profile. Therefore, our findings demonstrate the significant potential of this method to enhance our understanding of the global electron density distribution.

This description can be found in section 5 (Discussion) line 434-441.

Point 6

Second event has lightning source much farther from Arose footprint, however has greater whistler amplitude compared to the first event (according to comparison Figures 5 and 7).

Response 6

Lightning discharges produce a wide range of electromagnetic waves, including sferics and whistlers, which can travel through the Earth-ionosphere waveguide over long distances, as the attenuation rate for sferics is only around 2-3 dB per 1000 kilometers of propagation distance [33]. Sferics are low-frequency signals that can travel relatively far with little signal loss. The intensity of a lightning whistler signal is directly related to the strength of the lightning discharge that produced it. Stronger lightning discharges will produce more intense whistler signals, which can travel farther and be detected over longer distances than weaker signals. As a result, it is feasible to create the ray path in the second event, despite the fact that the lightning source is further away from the Arase footprint.

This description can be found in subsection 4.1 (Observed lightning whistler) line 290-294.

Point 7

Authors doesn’t introduce the approach to understand how the results could be validated. Only two events are considered and no comparison with experimental result or models are made. There is no enough statistics to see modification parameters distribution.

Response 7

As we mentioned in Response 1, the main point of the paper is to propose a method using the propagation characteristics of lightning whistlers. We demonstrate how to derive a more appropriate electron density in two different events. The results indicate that the electron density profiles for both events are adequately corrected toward the altitude. We realize that comparing our results with the other models and TEC measurements is necessary to evaluate the proposed method’s effectiveness and reliability. However, we must further examine and confirm the appropriate modification function, including the latitude and longitude, and thus, the quantitative accuracy evaluation is beyond this paper’s scope.

The description can be found in section 5 (Discussion) line 442-445.  

Point 8

Ray tracing is complicated procedure, while equation (1) and (2) give basis for it, the software implementation is not that simply. Please provide the reference to the code you are using for the ray tracing or reference to previous works where the implementation and validation is described.

Response 8

We add information about the reference of previous ray tracing program in subsection 3.1 (Calculation techniques) line 219. We developed the original ray tracing program introduced by Kimura and Goto [20] and Goto et al[9].

References :

[20] http://waves.is.t.kanazawa-u.ac.jp/

[9] Goto, Y., Kasahara, Y., Sato, T. Determination of plasmaspheric electron density profile by a stochastic approach. Radio Science, 2003, 38(3)

As we mentioned in subsection 2.2 (Magnetic field and electron density model) line 167-170 that the previous program used the simplest electron density model DE as a reference electron density profile. While the current work uses a combination of GCPM and IRI.

Goto et al. also used the original ray tracing program to propose a method for deriving a global electron density model using the propagation characteristics of Omega signals observed by the Akebono satellite [9] as mentioned in section 1 (Introduction) line 60.

In our electron density profile used in the ray tracing, we first calculate electron density and its derivatives in the (r,θ,Φ) coordinate system every 10o for longitude, 1o for latitude, and RE/10 for altitude. Then we derive the electron density and derivatives at the location of ray path by interpolating the neighboring grid data stored in the electron density model. As was discussed in Response 3, our current modification function (6) includes discontinuity in derivatives of electron density at r0 and rm, but it is negligible due to this interpolation in the ray tracing calculation.

Reviewer 3 Report (Previous Reviewer 4)

I am appreciated that the authors give the suitable reply to my questions. It can be accepted currently.

Author Response

We express our genuine appreciation to the reviewer for their valuable insights and productive recommendations after thoroughly reviewing our manuscript. We have revised the manuscript remotesensing- 2210927 entitled A proposal for modification of plasmaspheric electron density profiles using characteristics of lightning whistlers. We are sincerely hopeful that our revised manuscript will be received positively and accepted for publication in Remote Sensing.

Sincerely yours,

Desy Purnami Singgih Putri

This manuscript is a resubmission of an earlier submission. The following is a list of the peer review reports and author responses from that submission.

Round 1

Reviewer 1 Report

The Authors should clarify "several trial-and-error processes", written in Line 208. Is it any estimated optimization method?

The following items are minor comments.

* In Line 183, "at latitude 2.28deg and longitude 119.783deg" seems to be wrong. Maybe 52.28deg?

* "Equations 3" should be "Equation 3" in Lines 142 and 181.

* Ref. 7, "ekectron" should be "electron".

Reviewer 2 Report

This paper proposes a technique to reconstruct global electron density profiles using ray tracing by scrutinizing dispersion analyses of lightning whistlers.

Some important issues should be addressed.

1. The manuscript presents only one case for analysis. In Figure 5, the case of only 4 second is introduced. The data is too few to prove the effectiveness of the method proposed by the authors. The authors could present more data to convince the readers. The data obtained from different latitudes as well as in magnetic activity conditions could be presented to make the method more convincing.

2. The estimated electron density profile could be compared with the measured density by radio system or on-site satellites. The accuracy of the estimated data is still unclear, and the readers do not know how about the deviation between the estimated values and the actual values. It is necessary for the authors to provide the evidence for the accuracy of the method.

3. The English wording and grammar of the manuscript are very poor and should be improved. Many sentences that should be rewritten with the help of an English language expert.

Specific comments

Line 2 &Line4: “global density profiles” should be “global electron density profiles”

Line 11: “we satisfied the dispersion characteristics”. Bad English, It may be we are satisfied with the dispersion characteristics”

Line 89-90: “We compared an observed and ray-traced spectrogram to derive a more reliable density profile because the derived density profiles are not always accurate”. Bad English, I do not understand.

-Line 152: “lower altitudes” should be “lower altitude”

-Line 238: “we approximated the global density profile”. Bad English, I do not understand.

There are a lot of similar grammar and English usage problems in the manuscript, and a small number of them are listed above.

Reviewer 3 Report

Authors propose method to derive “global” electron density profiles using whistler waves. In the text “electron density profiles” are used along with “global electron density profiles” which is confusing since “global” in case of electron density profile (which is vertical distribution) is rather refereed to “all latitude and longitude”. It is suggested to go through the text and fix all “electron density profiles” with rather ionospheric electron density profiles or plasmashperic electron density profiles or combined profiles.

No discussion is provided on how electron density profile in the magnetosphere should look like. Proposed method is based on the modification of existing electron density profile as defined by equations (5). The electron density profiles is derived from GCPM and IRI. There many researches that study empirical models of the plasmaspheric electron density. They provide simple formula that could be modified without equations (5) which physical justification is not clear.

The step by step algorithm to achieve the results is not provided. Authors state that “after several trial-and-error processes, we set three parameters for Equation 4...”. Is it done manually? Can one reuse it for different event or every measurements should be handled manually.

Ray tracing which is referred briefly is an essential part of the research and is much more complicated than it is discussed in the paper [Chen, L., Bortnik, J., Thorne, R. M., Horne, R. B., & Jordanova, V. K. (2009). Three‐dimensional ray tracing of VLF waves in a magnetospheric environment containing a plasmaspheric plume. Geophysical Research Letters, 36(22)]. It is not evident from the paper that authors could really perform this procedure.

The experiment contains single event and the justification is made by delay time of the signal which is integral value. No comparison with existing models of the plasmasphere electron content or experimental data are made. Figure 8 indicates that generated density profile at longitude 120 and latitude 9. Assuming it for the vertical direction it is possible to do to make comparison to the plasmaspheric electron content derived using GNSS and ionosonde data (like in Yasyukevich A. S. et al. Correlation between Total and Plasmasphere Electron Content and Indexes of Solar and Geomagnetic Activity //2019 Russian Open Conference on Radio Wave Propagation (RWP). – IEEE, 2019. – Т. 1. – С. 87-90.) or recalculate from empirical model ( Ozhogin, P., Tu, J., Song, P., & Reinisch, B. W. (2012). Field‐aligned distribution of the plasmaspheric electron density: An empirical model derived from the IMAGE RPI measurements. Journal of Geophysical Research: Space Physics, 117(A6))

No analysis and discussion are made of applicability of the method. Namely how often Arase satellite footprints are in the right position with lightnings for the method to work.

Line 22: “down to the local maximum”

This is confusing since vertical sounding gives electron density up to ionospheric maximum for both ground based and topside sounding. If the local means “for given location” then it should be rewritten.

Line 23: “Another measurement technique is the use of the Global Positioning System (GPS) dual-frequency radio signal”

There are already 4 different global navigation satellite systems (GNSS) and GPS is only one of them. Researches uses signals of all of them. Ionospheric maps, databases and interactive systems were developed to provide access to those data.

Hernández-Pajares, M., Juan, J. M., Sanz, J., Orus, R., Garcia-Rigo, A., Feltens, J., ... & Krankowski, A. (2009). The IGS VTEC maps: a reliable source of ionospheric information since 1998. Journal of Geodesy, 83(3), 263-275.

Rideout, W., & Coster, A. (2006). Automated GPS processing for global total electron content data. GPS solutions, 10(3), 219-228.

Yasyukevich, Y. V., Kiselev, A. V., Zhivetiev, I. V., Edemskiy, I. K., Syrovatskii, S. V., Maletckii, B. M., & Vesnin, A. M. (2020). SIMuRG: System for ionosphere monitoring and research from GNSS. GPS Solutions, 24(3), 1-12.

Line 30-31: “One of these remote sensing techniques is radio sounding, which was used by IMAGE, the Imager for Magnetopause-to-Aurora Global Exploration satellite”

Please provide proper citation for this instrument.

Reinisch, B. W., Haines, D. M., Bibl, K., Cheney, G. G. I. A., Galkin, I. A., Huang, X., ... & Reiff, P. (2000). The Radio Plasma Imager investigation on the IMAGE spacecraft. In The IMAGE Mission (pp. 319-359). Springer, Dordrecht.

Reinisch, B. W., X. Huang, P. Song, G. S. Sales, S. F. Fung, J. L. Green, D. L. Gallagher, and V. M. Vasyliunas (2001), Plasma density distribution along the magnetospheric field: RPI observations from IMAGE, Geophys. Res. Lett., 28, 4521– 4524,

Fix typo in Reference 7 “ekectron” → electron

Line 1: global density profiles → ? global plasma/electron density profiles

Reviewer 4 Report

This paper proposed a method for generating global density profiles using ray tracing by scrutinizing dispersion analyses of lightning whistlers. They applied their method to a lightning whistler event on August 14, 2017. This is a general good paper to how the density modification affected the ray path dispersion characteristics. The reviewer have some minor comments.

Figure 5 is not clear and complete. For example, the legend label, y-axis label .... It should be re-plotted.

The language of this paper should be improved.